# Combined Feature Extraction and Random Forest for Laser Self-Mixing Vibration Measurement without Determining Feedback Intensity

**DOI:** 10.3390/s22166171

**Published:** 2022-08-18

**Authors:** Hongwei Liang, Minghu Chen, Chunlei Jiang, Lingling Kan, Keyong Shao

**Affiliations:** School of Electrical Information Engineering, Northeast Petroleum University, Daqing 163318, China

**Keywords:** vibration measurement, self-mixing interference, feature extraction, random forest

## Abstract

To measure the vibration of a target by laser self-mixing interference (SMI), we propose a method that combines feature extraction and random forest (RF) without determining the feedback strength (*C*). First, the temporal, spectral, and statistical features of the SMI signal are extracted to characterize the original SMI signal. Secondly, these interpretable features are fed into the pretrained RF model to directly predict the amplitude and frequency (*A* and *f*) of the vibrating target, recovering the periodic vibration of the target. The results show that the combination of RF and feature extraction yields a fit of more than 0.94 for simple and quick measurement of *A* and *f* of unsmooth planar vibrations, regardless of the feedback intensity and the misalignment of the retromirror. Without a complex optical stage, this method can quickly recover arbitrary periodic vibrations from SMI signals without *C*, which provides a novel method for quickly implementing vibration measurements.

## 1. Introduction

The laser self-mixing interference (SMI) technique [1,2,3] is an important and advanced method for ultra-precision measurements. Many interesting and important fields are inseparable from it, such as human ECG signal detection [4]; spectrum measurement [5,6]; and the measurement of phase, displacement, and vibration [1,7,8,9]. Because the SMI signal can reliably reflect the microvibration of an object, by analyzing the continuous SMI signal, the amplitude and frequency (*A* and *f*) of the microvibration of an object can be obtained.

In the field of microvibration measurements, it is important to recover information regarding the vibration of the target [8,10]. Phase expansion and modulation methods are often introduced into SMI vibration measurements. A simplified phase modulation and demodulation scheme was used to convert the vibration of the target into phase information, which improved the resolution of SMI non-contact measurements [11,12]. After determining *C*, Saqib Amin et al. proposed a method for power spectrum analysis of the SMI signal. They simplified the optics and efficiently measured the raw vibrations under various feedback mechanisms [13]. However, all of the above methods call for the demodulation of the real-time phase of the SMI in advance, and the linewidth enhancement factor (α) and feedback strength must be calculated. Wang et al. used phase deconvolution to determine the value of *C* and reconstructed the amplitude of a target under quadratic feedback [14,15]. Zhang et al. used multiple feedback techniques in SMI to improve the resolution and accuracy of displacement reconstruction [16]. However, these methods require complex experimental setup. In addition, Huang et al. used the FFT to expand the spectrum of the SMI signal into a Bessel function to determine the parameters of the vibrating target in sinusoidal form based on the order of the primary harmonics [17]. However, this method, based on FFT, requires a complex analysis of the spectrum of the SMI signal each time and various linear estimates of parameters, such as the main harmonic order.

In recent years, artificial intelligence techniques have been used to process SMI signals [18]. Imran Ahmed et al. used a generative adversarial network (GAN) to enhance the SMI signal corrupted by various noises. They performed noise removal and waveform enhancement of SMI signals under different noise conditions [19,20]. Kou et al. used an artificial neural network (ANN) for the discrimination of stripe tilt direction after detecting the number of stripes [21]. However, the number of stripes identified using this method was limited. Barland et al. built an SMI displacement dataset using periodic vibration signals and performed displacement reconstruction using a CNN [22]. However, this method requires a complex neural network structure, and the tuning of hyperparameters is time-consuming. The methods mentioned above also need to be analyzed when *C* and α are known. For time-series analysis [18], many applications using random forest (RF) for forecasting have also been included [23,24]. Time series analysis based on short time windows also facilitates subsequent model building. For example, Daqing Wu and Chenxiang Wu designed a method to calculate the traveling time of vehicles on roads across time periods [25]. They used various optimization objectives by considering the time windows of various parameters that change within a time window. In addition, feature extraction has been specifically applied in image classification algorithms. Huayue Chen et al. performed local feature extraction and used an extreme learning machine model to classify various spectral images [26]. Although classification algorithms for 2D images tend to be mature, feature extraction can also be applied to 1D SMI signals. Typically, this is a time-consuming and complex task. For example, Wang et al. extracted the features of a pulse waveform based on the reconstruction of the waveform by combining the inversion points of the SMI signal [27]. In contrast to the aforementioned time-consuming methods of feature extraction, we refer to the feature extraction algorithm (TSFEL) developed by Barandas et al. to extract features practically and quickly from the SMI signal [28,29], avoiding various nonlinear parameter estimations, simplifying the analysis of SMI signal without *C*, and directly extracting features of SMI signals to facilitate the construction of a regression model that can quickly measure the value of the target.

In this study, we propose a method that combines feature extraction and random forest for vibration measurements in SMI. We extracted 367 features with strong interpretation from the original SMI signal and normalized these features into a 0–1 interval as a one-dimensional vector fed into the RF model. With the optimal depth and number of nodes assigned to the decision tree, this feature vector is fed into the pretrained RF to predict *A* and *f* of the vibrating signal. The algorithm departs from the traditional idea of vibration measurement by SMI and recovers the original vibration directly from the SMI signal without determining *C* in advance or estimating other complex parameters.

The contributions of this study can be summarized as follows:(1).The proposed algorithm for vibration measurement by laser SMI does not require determination of the value of feedback intensity or estimation of the various parameters;(2).The feature extraction technique for SMI signals reduces the dimensionality of the signal and improves the universality of vibration measurements without involving complex calculations and analysis; and(3).Based on a large SMI dataset, the machine learning technique used in laser SMI with random forest makes the predicted amplitude and frequency of vibration coincide almost perfectly with the real vibration.

## 2. Theory of Laser SMI

The light output from the laser diode (LD) is reflected when it hits an external vibration. This part of the feedback light carrying information about the external object is returned to the resonant cavity of the laser, which then interferes with the laser inside the cavity, thereby modulating the output characteristics of the laser and thus generating the SMI signal. The circuit that forms the SMI is illustrated in Figure 1.

Combining the *Lang-Kobayashi* equation [30,31] and other theories [1,32,33], the phase and output power equations of SMI are expressed as follows:(1)$$\phi_{0}\left(t\right)=\phi_{F}\left(t\right)+C\sin\left[\phi_{F}\left(t\right)+\tan^{-1}\alpha \right]$$
(2)$$P_{F}\left(t\right)=P_{0}\left[1+m\cos\left(\phi_{F}\left(t\right)\right)\right]$$

The power ($P_{F}\left(t\right)$) of the SMI is related to the phase of the feedback light, and $P_{0}$ denotes laser power without feedback. In SMI, *C* is defined as:(3)$$C=\frac{\tau_{ext}}{\tau_{t}}k_{ext}\sqrt{1+\alpha^{2}}$$
where $\tau_{t}$ is the round-trip time for light in the laser cavity; $k_{ext}$ is the coupling coefficient that depends on the reflectivity of the exit laser facet and the reflectivity of the target; $\tau_{ext}$ is the round-trip time for light in the external laser cavity formed by the remote vibrating target; $L\left(t\right)$ is the length of the outer cavity, which varies with the vibrating target; and $\phi_{F}\left(t\right)$ and $\phi_{0}\left(t\right)$ represent the phases with and without light feedback, respectively. In Equations (4) and (5), $\phi_{0}\left(t\right)$ and $\phi_{F}\left(t\right)$ change linearly as $L\left(t\right)$ periodically varies, where $\lambda_{0}$ is the initial optical wavelength, and $\lambda_{F}$ is the perturbed laser wavelength.
(4)$$\phi_{0}\left(t\right)=2\pi \frac{2L\left(t\right)}{\lambda_{0}}$$
(5)$$\phi_{F}\left(t\right)=2\pi \frac{2L\left(t\right)}{\lambda_{F}}$$

In Equation (1), the value of *C* is changed by continuously adjusting the reflectivity and the position of the object relative to the light source. The SMI is in a strong feedback state during C≥4.6, where the stripes are jagged, and hysteresis occurs. During 1<C<4.6, the SMI is in a medium-feedback state. During 0<C≤1, the SMI operates in a weak feedback state, and the SMI signal is almost sinusoidal [11,34]. Without the addition of optical components, we studied the SMI signal in the weak-feedback state.

At 0<C≤1, assuming that the amplitude of the periodic vibration signal is *A* and the frequency is *f*, the distance between LD and the target is denoted as:(6)$$L\left(t\right)=L_{0}+A\cos\left(2\pi ft\right)$$
where $L_{0}$ is the initial distance between LD and the moving object. Once *C* is determined, the phase equation of SMI is established. Thus, the real-time phase ($\phi_{0}\left(t\right)$ and $\phi_{F}\left(t\right)$) of the SMI can be determined by combining Equations (1) and (2). Equation (4) shows that $L\left(t\right)$ is determined in accordance with $\phi_{0}\left(t\right)$. However, according to Equations (1), (3) and (4), *C* constantly changes in the actual optical path, and different feedback intensities can affect the real-time phase of the SMI. In addition, because of the continuity of *A* and *f*, the number of stripes in a cycle in the SMI signal is not necessarily an integer multiple of the half wavelength, resulting in instability in vibration measurement. Therefore, we used RF to quickly determine *A* and *f* of the vibrating target without adding optical instruments and considering the value of *C*.

## 3. Methods of Study

### 3.1. Feature Extraction of SMI Signal

As shown in Figure 2, we combined the physical characteristics of SMI [1] and time-frequency analysis to downscale the SMI signal and extracted 367 features from the temporal, spectral, and statistical domains as inputs to the RF.

First, in the temporal SMI waveform, the difference in the vibrating direction led to a left–right flip in the tilt direction of the stripe. This variation can be characterized in terms of temporal tilt, skewness, and kurtosis in statistics. From the perspective of the energy analysis and symmetry of the SMI signal, we also extracted other rich temporal features and statistical characteristics. Secondly, although *C* does not need to be calculated, the spectrum of the SMI signal after the Fourier transform reflects some details of the vibration waveform. This spectrum is concentrated in an interval that not only contains the phase variation of SMI but also accommodates many valid and interesting features for analysis of SMI signals, such as the power bandwidth, spectral centroid, and entropy. Combined with the time-frequency analysis, we also observe undulating changes in frequency in different Fourier short-time windows, where the darker part corresponds to the main component of spectrum in temporal. These features can also be extracted as input to the RF. In addition, some features do not have a single coefficient, and multiple values exist, which are more interpretable in a physical sense than the original SMI signal. Without performing complex signal analysis and parameter estimation on the SMI signal, the extracted features are detailed in Table 1 [28,29]. Where s represents SMI signal and the length of SMI signal is *L*; Δs is equal to $\left\{s_{1}-s_{0},s_{2}-s_{1},\cdots s_{L-1}-s_{L-2}\right\}$.

### 3.2. Vibration Measurement Algorithm Based on Random Forest

Dimensionality reduction of the SMI signal by feature extraction improves the efficiency of model training and the accuracy of actual vibration measurements. Therefore, we propose an algorithm to accurately measure the amplitude and frequency of microvibrations with RF, as shown in Figure 3.

On the left half of Figure 3, we randomly selected a frame of the SMI signal for feature extraction. Inevitably, the values of the extracted features were not at the same quantization level. Features with smaller quantization units increase the dependence of the model on the unit of measurement. To avoid over-reliance on the unit of measurement and improve the convergence speed of the algorithm, all features must be normalized. We normalized the features of the extracted SMI signals in the interval [0, 1] without any pre-estimation of intermediate parameters and calculating the feedback intensity and linewidth factor, which improves the computational efficiency of the algorithm. The law of normalization is as follows:(7)$$x_{i,j}^{\prime }=\frac{x_{i,j}-\min_{j}}{\max_{j}-\min_{j}},i=1,2,\cdots ,N,j=0,1,\cdots ,366$$
where $x_{i,j}$ is the value of the *j*th feature; $\max_{j}$ and $\min_{j}$ are the maximum and minimum values of the *j*th feature over *N* training samples, respectively; and $x_{i,j}^{\prime }$ is a value that falls within the interval [0, 1] after normalization. After normalization, these features were converted into a one-dimensional vector, which became the input of the RF model.

The RF regressor combines multiple decision trees by integration, sums the predictions of each tree, and averages them to obtain the best predictions of *A* and *f* of the vibrating target [35,36]. In addition, the robustness and stability of the RF were not destroyed by dynamic changes in *C*. The random forest integrates multiple classification and regressor trees (CART) with weak learning ability to form a powerful network to measure the *A* and *f* of the target vibration. The algorithm for the RF regression is shown on the right-hand side of Figure 3.

In the original SMI dataset, there are *N* samples, each of which is a feature vector of length *M*. *K* decision trees are constructed based on random sampling with put-back several times, and *K* new SMI sample sets are obtained. From a=1 to a=K, we repeated the following steps to train the *a*th decision tree:(i)When splitting the nodes of each decision tree, some features are intercepted from the subvectors of the SMI feature vector on the nodes. Furthermore, the number of features selected for each node must not exceed the value of *Max_feature*;(ii)Select an optimal subset of features to continue training, split the nodes into sub-nodes, and select the nodes with the best split points.(iii)Once the number of layers of a node reaches its *Max_depth*, the splitting stops.

In this manner, the *K* decision tree was trained. To predict *A* and *f* from the SMI signal, a one-dimensional SMI feature vector was fed into each decision tree for regression. The mean value of the prediction of each decision tree was considered as final measurement of *A* and *f* of the original periodic vibration recovered from the SMI signal. Disregarding the effects of the magnitudes of *A* and *f*, we used R2_score to observe the predictive effect of the model [37,38,39,40]. R2_score was calculated as follows:(8)$$R2\_score=1-\frac{\sum_{i=0}^{N-1}{\left(y_{i}^{A,f},{\hat{y}}_{i}^{A,f}\right)}^{2}}{\sum_{i=0}^{N-1}{\left(y_{i}^{A,f},{\bar{y}}_{i}^{A,f}\right)}^{2}}$$
where $y_{i}^{A,f}$ and ${\hat{y}}_{i}^{A,f}$ are the true and predicted values, respectively, corresponding to the SMI feature vector, $X=\left(x_{i}^{0},x_{i}^{1},\cdots ,x_{i}^{N-1}\right)$, for the *i*th input in the prediction of *N* samples. The mean of the true labels of *N* samples was ${\bar{y}}_{i}^{A,f}$.

### 3.3. Validation Analysis Based on Simulation Dataset

Combining Equations (1)–(4) and the simulation code of SMI summarized in [1,32], we simulated and tested the RF using *Python3.6*, setting the wavelength (λ) of the laser to 650 nm.

Without considering the dynamics of *C*, we extracted the features of SMI signals directly from the temporal, statistical, and spectral domains to form a large set of SMI simulation data. In this way, we obtained 48,000 samples in the range of 0~100 Hz for *f* and 0~2 µm for *A*. For each SMI signal, we used 1200 sampling points as a window for flexible feature extraction using TSFEL to obtain 367 features corresponding to the amplitude and frequency of the original vibrating signal.

As shown in Figure 4, the extracted feature vectors were normalized. The distribution of the features indicated that the spectral features strongly contributed to the measurement of *f*. During validation of the proposed algorithm using the simulation dataset, 30% of the dataset was reserved for testing.

Some of the predicted results from the simulated test data are presented in Figure 5. Although the RF can output continuous values of both *A* and *f*, we compared their predicted values separately. The blue points represent the *A* and *f* values predicted by the model, whereas the red bars represents the real *A* and *f* of the vibration. Owing to the richness of our dataset, the amplitude error measured by RF was, at most, λ/4, and the measurement error was lower in the vast majority of samples. For the measurement of *f*, the true and measured values were almost perfectly fitted. Combining the predicted results of *A* and *f*, the performance score of RF in the simulation test set was 0.98, which indicates that RF has a very high prediction capability for the SMI signal. We obtained a large amount of data from the optical platform and used this model to test and validate the experimental data in the next section.

## 4. Analysis of Experimental Results

### 4.1. SMI Experimental Dataset

To generate the SMI experimental dataset, we set λ of the laser diode to 650 nm and the sampling frequency of the acquisition card to 20 kHz. As shown in Figure 6, We used an acquisition card to transfer the acquired SMI signals to a *PC*. We extended the range of measurement by varying the vibration parameters of the vibrating source without estimating *C*. Thus, we built a large number of samples in the range of 0~8 µm for *A* and 5~25 Hz for *f*, forming the SMI experimental dataset. We collected 60,000 points for the SMI signal under each pair of parameters using 1200 sampling points and the corresponding *A* and *f* as a set of samples. These data were used for subsequent construction and training of the real RF.

### 4.2. Result Analysis

We extracted the features from the experimental signals and performed an analysis. As shown in Figure 7, the actual SMI signal was entrained with a significant amount of noise and contained many burrs. We used wavelet transform [41] to filter the SMI signal such that the curve was smoothed. Next, using Fourier transform and STFT, the spectrum and time-frequency maps were obtained. After Fourier transform, the spectrum of the SMI signal was entrained with noise, which could not be completely eliminated. However, the spectrum maps are also more concentrated in one interval, similar to the spectrum of the simulated signal, which does not affect the extraction of the basic features. After STFT, many features can be extracted from the short-time spectrum within each window function.

As shown in Figure 8, among the 367 extracted features, dense spectral features contributed the most to the original vibration. Although certain temporal features have higher scores, these high-scoring features are fewer and less widely distributed than the spectral features.

In addition, the contribution of the spectral features to the recovery of the original vibration was within a stable range, which helped maintain the predictive effect of the RF. This also indicates that the prediction accuracy of the model was not affected without considering the changes in *C*. The statistical features scored low, but there were some high-scoring features. Although the contribution of all features to the actual vibration from the SMI signal obtained from the optical platform is slightly lower than that of the simulated data, these features are essential. We also analyzed the performance of the experimental dataset compared with the simulated dataset, adjusting the various parameters of each decision tree in the RF.

In Figure 9a, when Max_depth<10 is applied, the RF appears to be overfitted. As the depth of each decision tree continued to increase, the prediction performance of RF steadily improved. Compared with the simulation dataset, the experimental data were affected by the physical environment and contained more noise. After filtering out the noise, although the prediction score was not as high as that of the simulation data, RF achieved an accuracy of more than 94% on the experimental data during Max_depth≥10. In addition, we analyzed the prediction effect of various *Max_features*. In Figure 9b, at each node of the decision tree in the forest, we found an optimal subset of features that resulted in the highest accuracy of the RF. Near Max_feature=25, R2_score starts to enter the flat state. This implies that the minimal length of this feature subset is approximately 50 when the RF achieves optimal performance.

Using the final model trained with the above optimal parameters, we reacquired SMI signals from the optical platform for testing and prediction. As shown in Figure 10, we adjusted various parameters of the vibration source to obtain different SMI signals. On the one hand, for the same oscillation pattern of the target, we randomly picked different starting points from the corresponding SMI signal and intercepted different sampling sequences as part of the test data. On the other hand, we also selected the SMI signals under different oscillation laws as another part of the test data. Using RF, we smoothly predicted the *A* and *f* of the original vibration corresponding to these SMI signals.

The prediction accuracy on the actual SMI signal is 94.75%. The predicted *A* and *f* almost perfectly coincided with the real vibration. The predicted values at some points deviated from the true values due to the presence of noise and abrupt changes in the shape of the stripes. However, this deviation gradually decreased after several measurements.

## 5. Discussion

The aim of this study was to perform vibration measurements in SMI utilizing feature extraction and random forest without determining *C* and to improve the interpretability and practicality of the algorithm.

In addition to RF, lasso and ridge have been studied in various fields [42]. For SMI vibration measurements, we compared each of these algorithms.

Although ridge does not rely on a large number of decision trees for integrated learning as RF does, its performance relies heavily on the change in the regularization coefficient (γ) and the choice of gradient solver. We used multiple cross-validation methods to continuously modify the value of γ in the range of 0–1 and evaluated the effect of each pretrained model on the experimental SMI test data. In Figure 11, to solve for the minimum of the loss function of the ridge regressor, we present a plot of R2_score with γ under different gradient descent schemes.

The optimal solution of the ridge without considering *C* also differs depending on the gradient descent rules. In the SMI training data, we found that saga and sag reached local convergence ahead of time. However, the weight of the ridge derived from this convergence point is not globally optimal. As γ increases, the two laws, *saga* and *sag*, are the least effective for measuring *A* and *f* against actual vibration. In contrast, the remaining three gradient solvers are dedicated to determining the global optimal solution of the ridge through dynamic search [43,44]. The *cholesky* solver assigned the best weights of the ridge to *367* features, achieving the highest performance at γ=0.05, where the accuracy of the measurement was 77.41%.

Furthermore, when analyzing the robustness of various algorithms, we varied the length of the SMI signal to be subjected to feature extraction. Under the optimal rule of gradient solution, we continuously adjusted γ to analyze the effects of vibration measurements under different lengths of SMI signal.

In Figure 12, at γ≤0.002, the ability of either ridge or lasso to measure the *A* and *f* with the 367 extracted features increases as γ continues to increase. The degree of underfitting decreased rapidly at this stage, and the model converged faster. However, compared to ridge, the coefficients of lasso can be adjusted in a very narrow range, which easily causes lasso to converge locally, resulting in the highest accuracy at a very low γ. However, as γ increases, the accuracy of lasso drops sharply, even to zero, exacerbating the instability of measuring the *A* and *f* of the actual vibration. From the analysis of the speed of convergence and stability of the measurement, ridge reached global convergence, although the value of γ at the highest performance occurred later than that of lasso. During γ>0.05, the accuracy decreases slowly, although the performance of the ridge continues to underfit as γ increases. This means that in addition to RF, ridge is also useful for any range of stable vibration measurements, not limited to existing *A* and *f*. Additionally, in Figure 12, we explored the effectiveness of these algorithms for measuring *A* and *f* with different lengths of SMI signals. Although the scoring curves of both algorithms were the lowest at a length of 1300 for the SMI signal, they could be improved by increasing the amount of training data. Overall, the accuracy of these algorithms for measuring *A* and *f* gradually increases as the length of the SMI signal increases. This suggests that adjusting the length of the SMI signal can significantly improve the measurements of *A* and *f* of the vibrating target using these algorithms.

As shown in Table 2, the accuracy of RF tends to be stable, regardless of the length of the SMI signal. The RF had the highest accuracy for the same length of SMI signal. Although the accuracy of the vibration measurement by ridge was 57.68% at a length of 1300, the lowest value in Table 2, the accuracy can be improved to more than 0.8, when the length of the SMI signal was changed several times. In Figure 12, the convergence of ridge has a somewhat wider range, which is better than that of lasso. The accuracy of the measurement of *A* and *f* by lasso was maintained between 0.66 and 0.73, which is difficult to improve. This is impractical for the measurement of a vibrating target using SMI. After feature extraction for SMI signals of arbitrary length, the RF had the highest accuracy for both *A* and *f* measurements. This shows that our proposed algorithm has no fixed limitation on the length of the SMI signal and is suitable for real-time vibration measurements.

After comparing the above algorithms, we found that our RF maintains the highest accuracy in both the simulation and experimental data, with the lowest measurement errors for *A* and *f*, making it more practical. The performance of lasso was the worst in terms of the experimental data. It has the lowest R2_score when predicting *A* and *f* of the actual vibration. In terms of stability, the ridge outperformed the lasso in dealing with SMI signals containing noise when measuring microvibrations. It may be beneficial to combine ridge with RF, considering the creation of an integrated model so that the final algorithm will be more robust.

When the reflector in the laser cavity is misaligned when the object is vibrating, the round-trip time for the laser to hit the vibrating object and return to the laser cavity directly affects the value of the feedback intensity. However, no matter how the feedback intensity changes, the generated self-mixed signal always reflects the *A* and *f* of the original vibrating target. Therefore, our method has no requirement for a fixed position of the reflector in the laser cavity and has a good tolerance for misalignment of the reflector.

The idea is to measure the original vibration accurately, even without considering the feedback strength. For unsmooth planar vibrations, noise is inevitably superimposed on the SMI signal. In our experiments, we chose a reflection plane of vibration similar to the reflection diffusion surface, with no fixed requirement for the reflection coefficient of the plane, enabling the feedback light to return to the resonant cavity of the laser and interfere with the original output light to form the SMI signal. After filtering out the noise, we successfully performed effective vibration measurements with the proposed method. Thus, the method is applicable to a vibrating reflective diffuse surface. Additionally, in some cases, the target vibrates at high frequencies in the kilohertz and megahertz range, placing considerable demands on the high-frequency sampling accuracy of the sampling instrument. However, our method can still perform feature extraction of the generated SMI signal, which reflects information such as amplitude and frequency of the vibrating object. With subsequent research, our method can be used to measure vibrations at high frequencies.

## 6. Conclusions

For vibration measurement by SMI, we proposed an algorithm for vibration measurement based on feature extraction of the SMI signal and RF without determining any feedback coefficients. This algorithm directly dimensionalizes the SMI signal, extracts the features of the SMI signal, and combines the RF to recover *A* and *f* of the original vibrating target. Compared to existing studies in vibration measurement by laser SMI, our experimental results show that this method is simple to operate, not involving complex optical platforms and calculation for phase unwrapping, and does not require the determination of any optical parameters, such as *C*. This method is tolerant to misalignment of the retromirror in the laser cavity and can be applied to a vibrating reflecting diffusing surface. Furthermore, the method has no fixed requirement for the selection of a laser [45,46]. If the threshold current is within the permissible range of the laser and the output light at a certain wavelength can produce an SMI signal, vibration measurement can be achieved [47]. Therefore, the method is inclusive of laser selection, convenient for various advanced laser materials for SMI and vibration measurement, and has a high degree of universality.

To construct the feature vector of the SMI signal, we extracted rich features from the spectral, statistical, and temporal domains as inputs to the RF. This eigenvector provides strong interpretability for vibration measurements using RF. When assessing the feature importance, each feature provides a stable contribution to the prediction performance of *A* and *f*. Furthermore, by comparing and analyzing multiple algorithms, we verified that the extracted features of the SMI signal are best suited for vibration measurements using RF.

In summary, relying on a large amount of SMI data, this algorithm for vibration measurement can be a valuable element for various types of laser sensing devices. Although the increased amount of data and good preprocessing efforts help to improve the measurement accuracy, it requires a lot of time to collect SMI signals from optical platforms to further investigate the correspondence between the SMI signals and the original vibration. Therefore, we emphasize that machine learning algorithms incorporating SMI for vibration measurements rarely have hard and fast rules, and these designs are often a research area of their own. Our next step will be to combine the theory of laser SMI to improve machine learning methods so that only a limited number of training samples are needed to be equally useful for a wide range of vibration measurements.

## Figures and Tables

**Figure 1 sensors-22-06171-f001:**
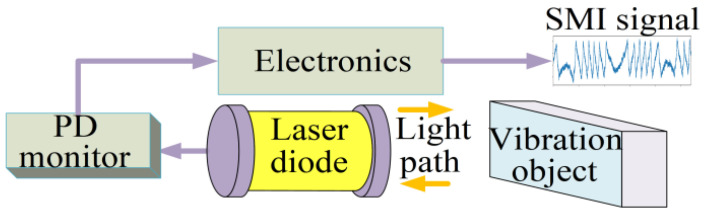
Circuit for forming SMI.

**Figure 2 sensors-22-06171-f002:**
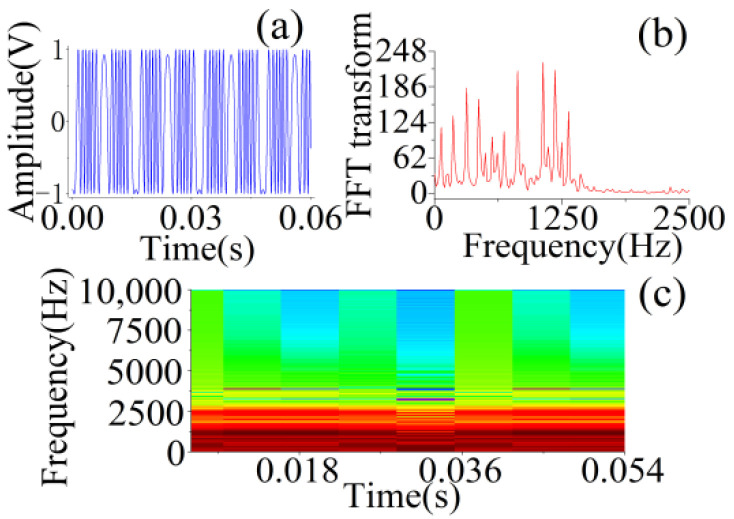
Feature extraction in different aspects of the SMI signal. (**a**) Temporal signal. Sampling rate: 20 kHz, *A* = 1.05 μm, *f* = 62.5 Hz, *C* = 0.1. (**b**) Spectrogram of the SMI signal. (**c**) SMI STFT analysis plot. Hamming window; window length: 256; overlap points: 128.

**Figure 3 sensors-22-06171-f003:**
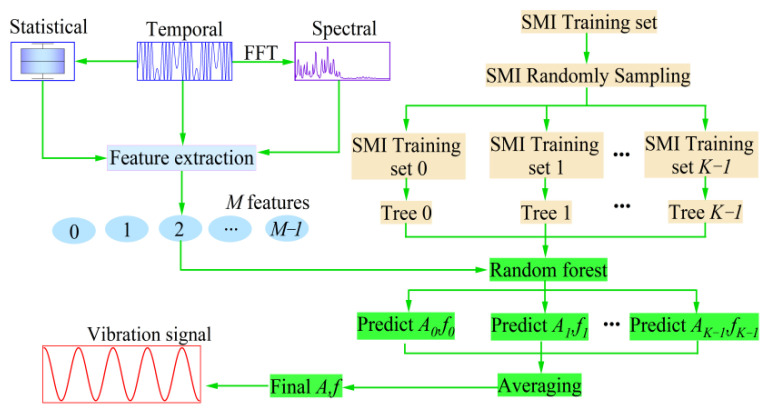
Vibration measurement framework combining RF and SMI.

**Figure 4 sensors-22-06171-f004:**
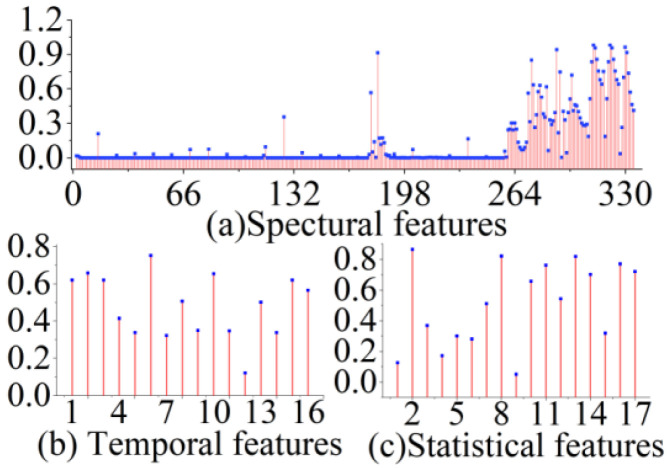
Extracted features from different aspects after normalization. (**a**) 334 spectral features, (**b**) 16 temporal features, and (**c**) 17 statistical features.

**Figure 5 sensors-22-06171-f005:**
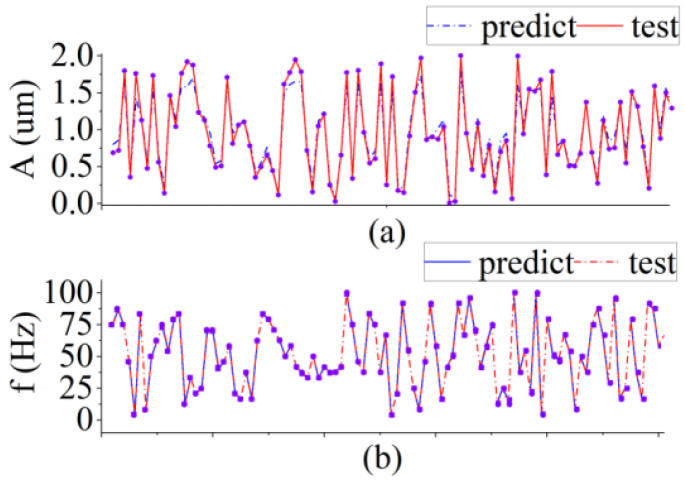
Prediction of *A* and *f* in the simulation test. (**a**) Measurement of *A* in the range of 0~2 µm. (**b**) Measurement of *f* in the range of 0~100 Hz.

**Figure 6 sensors-22-06171-f006:**
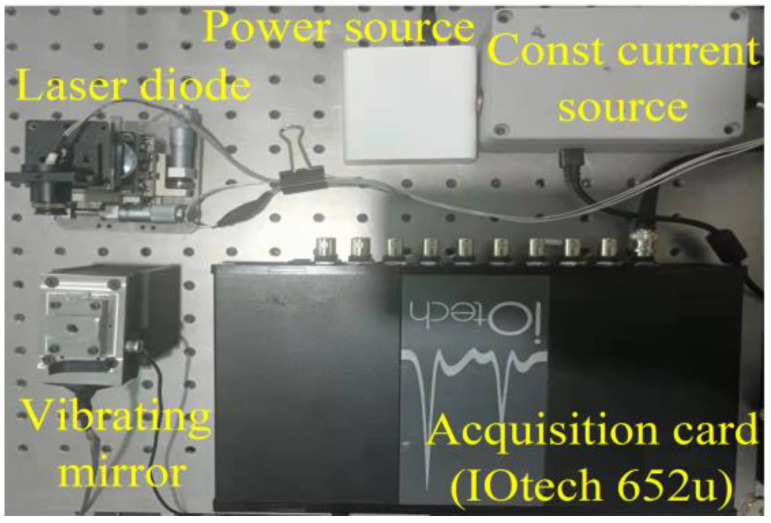
Platform for acquiring SMI microvibration data.

**Figure 7 sensors-22-06171-f007:**
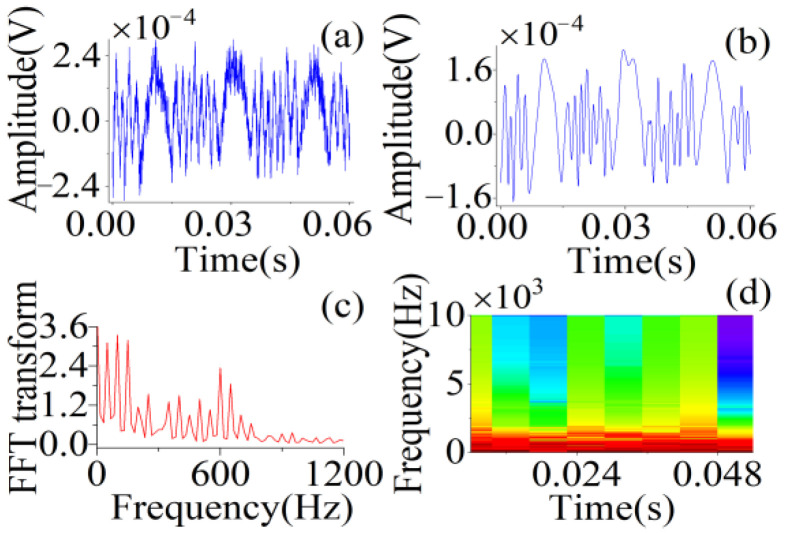
(**a**) SMI signal obtained with an optical platform with *A* = 2.075 µm and *f* = 25 Hz. (**b**) The waveform of an actual SMI signal with wavelet transform. (**c**) Fourier transform of SMI signal after wavelet transform. (**d**) STFT of SMI signal. Hamming window; window points: 256; overlapping points: 128.

**Figure 8 sensors-22-06171-f008:**
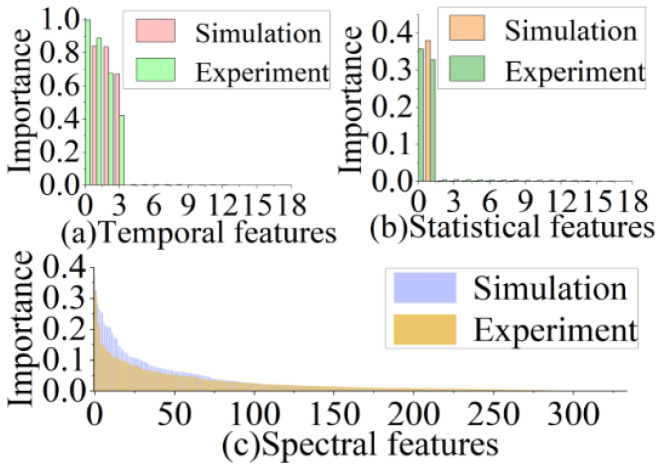
Temporal, statistical, and spectral feature importance analysis on simulated and experimental dataset.

**Figure 9 sensors-22-06171-f009:**
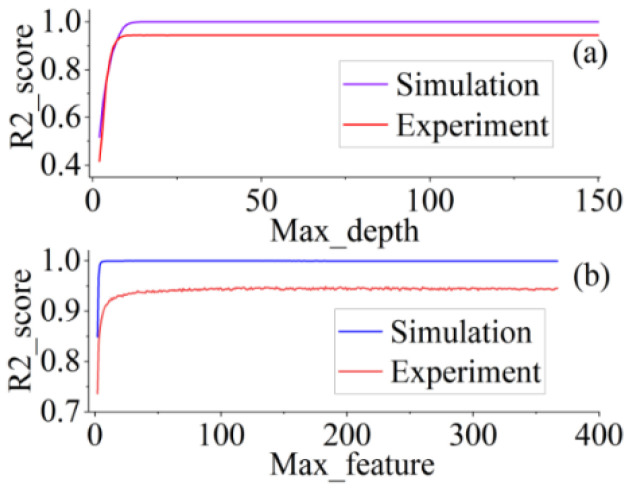
Performance curves under various parameters of RF. (**a**) R2_score with different *Max_depth*. (**b**) R2_score with different *Max_feature*.

**Figure 10 sensors-22-06171-f010:**
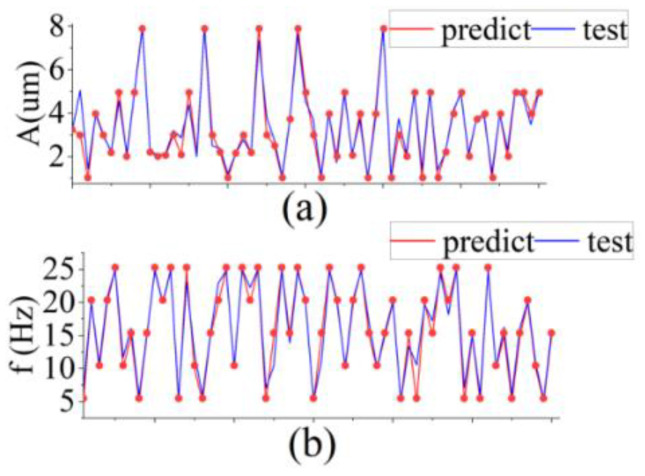
Vibration measurements on SMI experimental data. (**a**) Measurement of *A* in the range of 0~8 µm. (**b**) Measurement of *f* in the range of 5~25 Hz.

**Figure 11 sensors-22-06171-f011:**
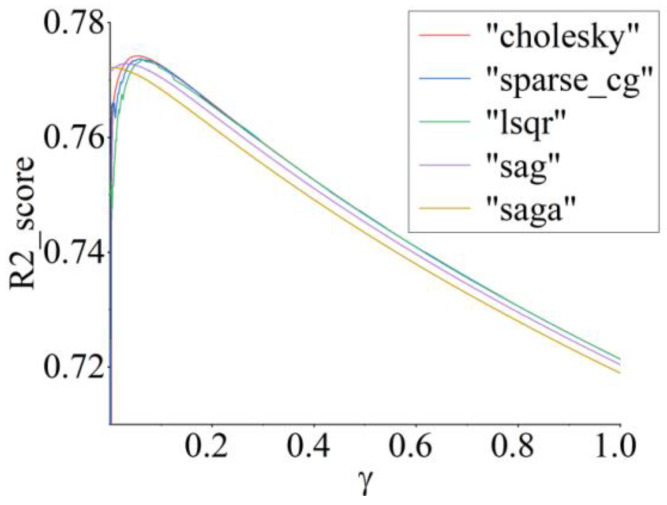
Joint analysis of gradient solver and γ for vibration measurement by SMI.

**Figure 12 sensors-22-06171-f012:**
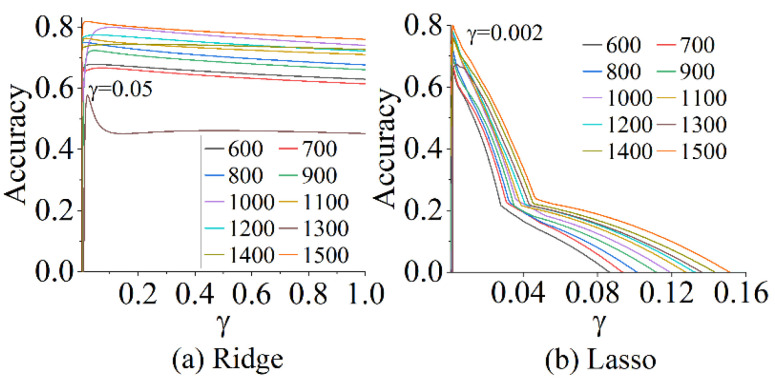
Curves of the accuracy of measuring *A* and *f* with γ under different lengths of SMI signals. (**a**) Accuracy changes with γ on ridge. (**b**) Accuracy changes with γ on lasso.

**Table 1 sensors-22-06171-t001:** List of 367 values of extracted features.

Temporal Features		
Name	Definition	Number of Values
Absolute energy	Integration of the square of the voltage of the SMI signal	1
Total energy	Total energy in a frame of SMI signal: $\sum_{i=0}^{L-1}s_{i}^{2}/\left(t_{L-1}-t_{0}\right)$	1
Mean absolute diff	Mean absolute differences of SMI signal: mean ($\left|\Delta s\right|$)	1
Median absolute diff	Median absolute differences of SMI signal: median ($\left|\Delta s\right|$)	1
Signal distance	Total distance traveled by SMI signal	1
Signal slope	The degree of inclination of the stripes of SMI and the direction of the original vibration	1
Positive turning points	The number of points where the signal starts to rise from the trough point	1
Negative turning points	The number of points where the signal starts to fall from the peak point	1
Mean diff	Mean of differences of SMI signal: mean $\left(\Delta s\right)$	1
Median diff	Median of differences of SMI signal: median $\left(\Delta s\right)$	1
Neighborhood peaks	The number of peaks from a defined neighbourhood of SMI signalm	1
Autocorrelation	Autocorrelation of the SMI signal	1
Centroid	The centroid of the SMI waveform along the time axis	1
Area under the curve	Computes the area under the waveform of the SMI signal with the trapezoid rule	1
Sum absolute diff	Sum of absolute differences of SMI signal: $\sum_{i=0}^{L-1}\left|\Delta s_{i}\right|$	1
Zero crossing rate	The total number of times that the SMI signal changes from positive to negative or vice versa	1
**Statistical Features**		
**Name**	**Definition**	**Number of Values**
ECDF percentile	Computes the values of empirical cumulative distribution function along the time axis	2
Histogram	The values of histograms of the SMI signal	5
Interquartile range	Computes the interquartile range of the data points of the SMI signal	1
Root mean square	Square root of the arithmetic mean (average) of the squares of original signal	1
Standard deviation	Standard deviation of the SMI signal	1
Median absolute deviation	Median absolute deviation of the SMI signal: $median\left(\left|s-median\left(s\right)\right|\right)$	1
Mean absolute deviation	$\sum_{i=0}^{L-1}\left|s_{i}^{2}-mean\left(s\right)\right|/L$	1
Variance	$mean{\left(\left|s-mean\left(s\right)\right|\right)}^{2}$	1
Mean	Mean value of the SMI signal	1
Median	Median value of the SMI signal	1
Kurtosis	Describes the steepness of the pattern of all fetched values in the SMI signal	1
Skewness	Describes the symmetry of the waveform of the SMI signal	1
**Statistical Features**		
**Name**	**Definition**	**Number of Values**
Spectral kurtosis	Measures the flatness of the spectrum around the mean value of the SMI signal.	1
Spectral variation	Computes the amount of variation of the spectrum over time	1
Spectral slope	Computed using a linear regression over the spectral amplitude values	1
Spectral maximum	Maxium frequency of the SMI signal	1
Spectral median	Median frequency of the SMI signal	1
Spectral entropy	Normalized value of spectral entropy of the SMI signal based on Fourier transform	1
Fundamental frequency	Explains the content of the signal spectrum	1
Spectral roll-off	The frequency at which 95% of the signal magnitude is contained	1
Spectral skewness	Measure of the flatness of the spectrum around its mean value	1
Spectral spread	The spread of the spectrum around its mean value	1
Positive turning points	The number of positive turning points of the fft magnitude signal	1
Fft mean coefficient	Computes the mean value of each spectrogram frequency	256
Max power spectrum	Computes the maxium power spectrum density of the SMI signal	1
Spectral centroid	The spectral center of gravity	1
Decrease	The decreasing in the spectral amplitude	1
Spectral distance	Distance of cumulative sum for the SMI signal of FFT elements to the respective regression	1
Wavelet absolute mean	The discrete wavelet transform (CWT) absolute mean value of each wavelet scale	9
Wavelet standard deviation	The CWT standard deviation of each wavelet scale	9
LPCC	The linear prediction cepstral coefficients	13
MFCC	Mel frequency cepstral coefficients, which provide the power information	12
Power bandwidth	Power spectrum density bandwidth of the SMI signal	1
Wavelet energy	The CWT energy of each wavelet scale	9
Wavelet variance	The CWT variance value of each wavelet wavelet scale	9
Wavelet entropy	The CWT entropy of the SMI signal	1

**Table 2 sensors-22-06171-t002:** Accuracy of different lengths of SMI signals for the measurement of *A* and *f*.

Frame	600	700	800	900	1000
RF	88.95%	88.37%	92.92%	90.27%	94.63%
Lasso	66.58%	67.84%	72.67%	70.03%	78%
Ridge	67.86%	67.35%	75.04%	72.67%	80%
**Frame**	**1100**	**1200**	**1300**	**1400**	**1500**
RF	93.15%	94.75%	92.78%	93.65%	94.63%
Lasso	66.58%	66.58%	67.84%	72.67%	70.03%
Ridge	76.39%	77.41%	57.68%	74.42%	81.84%

## Data Availability

The data presented in this study are available on request from the corresponding author. The data are not publicly available due to privacy or ethical restrictions.

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
