# Peer review of "Combined Feature Extraction and Random Forest for Laser Self-Mixing Vibration Measurement without Determining Feedback Intensity"

_sensors, 2022, doi:10.3390/s22166171_

Round 1
Reviewer 1 Report
There is little content, which need be revised according to the comment of reviewer in order to meet the requirements of publish. A number of concerns listed as follows:
(1) The abstract should be rewritten to reflect the significance of the proposed work. The current abstract shows a lot of background information.
(2) Please highlight your contributions in introduction.
(3) To explore Comparative results with existing approaches/methods relating to the proposed work.
(4) The authors need to interpret the meanings of the variables.
(5) In page 3, “First, in the temporal SMI waveform…..” ——>” Firstly, in the temporal SMI waveform, …”, ……..In page 4, “Second, a…”——>” Secondly, …”, …….
(6) In Table 1., 1+1, 2+5,1+1+1, 256+1,….. Please explain what they mean?
(7) Conclusion: What are the advantages and disadvantages of this study compared to the existing studies in this area?
(8) The inspiration of your work must further be highlighted. Some suggested recent literatures should add. For example,
[1]https://doi.org/10.3390/agriculture12060793
[2]https://doi.org/10.1109/JSTARS.2021.3059451
[3]https://doi.org/10.1016/j.engappai.2022.105139
[4]https://doi.org/10.1007/s10489-022-03719-6
(9) Correct typological mistakes and mathematical errors.
Reviewer 2 Report
The authors present a novel method of measurement of the parameters of a vibrating structure using selfmixing interferometry in a laser diode cavity. The most interesting aspect concern the use of feature extraction with the Random Forest - RF method. The overall quality of the paper is good and experimental results confirm an accurate agreement in the measurements of amplitude and frequency of vibration of the structure with the RF algorithm. I note few questions to consider in the final version of the mauscript :
- It is not clearly detailed the feedback value C> 4.6 . Explain the origin of this particular value of C with strong feedback issued from relations 1 - 2
- Is the method very tolerant to a misalignement of the retromirror in the laser cavity. Also does the method apply to a vibrating reflecting diffusing surface ?
- The laser used is a semiconductor laser. Which change of the parameters to consider when using another type of laser , ie a diode pumped solid state laser.
- There are conditions where the structure may be vibrating at higher frequencies like kHz- MHz range. Is a well suited method to operate in this frequency range. Discuss limitations in term of performances.
To conclude the paper is of interest for the accurate metrology of vibrating structures. The RF algorithm demonstrates significant progress both for the optical implementation and for the signal processing. The manuscript can be published in the journal after taking account of the above comments and questions.
Round 2
Reviewer 1 Report
I have appreciated the deep revision of the contents and the present form of this manuscript. All my previous concerns have been accurately addressed. I think that this paper can be accepted.